# National Colorectal Cancer Screening Program in Lithuania: Description of the 5-Year Performance on Population Level

**DOI:** 10.3390/cancers13051129

**Published:** 2021-03-06

**Authors:** Audrius Dulskas, Tomas Poskus, Inga Kildusiene, Ausvydas Patasius, Rokas Stulpinas, Arvydas Laurinavičius, Laura Mašalaitė, Gabrielė Milaknytė, Ieva Stundienė, Lina Venceviciene, Kestutis Strupas, Narimantas E. Samalavicius, Giedre Smailyte

**Affiliations:** 1Laboratory of Clinical Oncology, National Cancer Institute, 1 Santariskiu Str., LT-08406 Vilnius, Lithuania; 2Institute of Clinical Medicine, Faculty of Medicine, Vilnius University, M. K. Čiurlionio Str. 21/27, LT-03101 Vilnius, Lithuania; tomas.poskus@santa.lt (T.P.); rokas.stulpinas@vpc.lt (R.S.); arvydas.laurinavicius@vpc.lt (A.L.); laura.masalaite@santa.lt (L.M.); gabriele.milaknyte@santa.lt (G.M.); ieva.stundiene@santa.lt (I.S.); kestutis.strupas@santa.lt (K.S.); narimantas.samalavicius@gmail.com (N.E.S.); 3Laboratory of Cancer Epidemiology, National Cancer Institute, LT-08406 Vilnius, Lithuania; inga.kildusiene@nvi.lt (I.K.); ausvydas.patasius@nvi.lt (A.P.); giedre.smailyte@nvi.lt (G.S.); 4Department of Public Health, Institute of Health Sciences, Faculty of Medicine, Vilnius University, LT-03101 Vilnius, Lithuania; 5National Centre of Pathology, Affiliate of Vilnius University Hospital Santaros Klinikos, 5 P. Baublio Str., LT-08406 Vilnius, Lithuania; 6Clinic of Internal Medicine, Family Medicine and Oncology, Faculty of Medicine, Vilnius University, LT-08410 Vilnius, Lithuania; lina.venceviciene@santa.lt; 7Centre of Family Medicine, Vilnius University Hospital Santaros Klinikos, LT-08410 Vilnius, Lithuania; 8Department of Surgery, Klaipeda University Hospital, 41 Liepojos Str., 92288 Klaipeda, Lithuania; 9Health Research and Innovation Science Center, Faculty of Health Sciences, Klaipeda University, 84 Herkaus Manto Str., LT-92288 Klaipeda, Lithuania

**Keywords:** colorectal cancer, colorectal cancer screening, national cohort

## Abstract

**Simple Summary:**

The first Lithuanian analysis of colorectal cancer screening program is presented in our manuscript. We found that program is run with minimal expenses and still surpasses minimal requirements proposed by the European Union. Still the coverage is lower being 49.6% and must be improved.

**Abstract:**

We aimed to report the results of the implementation of the National Colorectal Cancer (CRC) Screening Program covering all the country. The National Health Insurance Fund (NHIF) reimburses the institutions for performing each service; each procedure within the program has its own administrative code. Information about services provided within the program was retrieved from the database of NHIF starting from the 1 January 2014 to the 31 December 2018. Exact date and type of all provided services, test results, date and results of biopsy and histopathological examination were extracted together with the vital status at the end of follow-up, date of death and date of emigration when applicable for all men and women born between 1935 and 1968. Results were compared with the guidelines of the European Union for quality assurance in CRC screening and diagnosis. The screening uptake was 49.5% (754,061 patients) during study period. Participation rate varied from 16% to 18.1% per year and was higher among women than among men. Proportion of test-positive and test-negative results was similar during all the study period—8.7% and 91.3% annually. Between 9.2% and 13.5% of test-positive patients received a biopsy of which 52.3–61.8% were positive for colorectal adenoma and 4.6–7.3% for colorectal carcinoma. CRC detection rate among test-positive individuals varied between 0.93% and 1.28%. The colorectal cancer screening program in Lithuania coverage must be improved. A screening database is needed to systematically evaluate the impact and performance of the national CRC screening program and quality assurance within the program.

## 1. Introduction

According to the most recent GLOBOCAN data (2018) [1], colorectal cancer (CRC) is the fourth most common cancer worldwide, with an annual incidence of more than 1,800,000 cases and the third highest mortality rate [1]. It is the second most common cancer in Lithuania—1892 men and women are estimated to be diagnosed with colorectal cancer (11.4% of all cancer diagnoses) [2]. It carries the third-most burden of cancer in the country [3].

The adenoma–carcinoma sequence is clearly established in CRC [4] it takes at least 10 years to progress from adenoma to carcinoma. The removal of premalignant adenomas is believed to reduce the incidence of CRC. It is possible to screen for CRC due to a few factors: high incidence, long development course, and effective endoscopic treatment options in premalignant stage [5]. In addition, this is the reason why the European Union recommends the screening using fecal occult blood test (FOBT) or fecal immunochemical test (FIT) [6,7]. The quality of CRC screening program has been developed previously by an expert group [8,9].

The CRC screening program was initiated in Lithuania in June 2009. From January 2014 the program covers all the population aged 50–74 in Lithuania. The aim of the program is to reduce CRC mortality by detecting the disease earlier and, possibly, to reduce the incidence of CRC by removing advanced adenomas. We have previously published our initial results of the program showing the uptake rate of 46.0% over 3 years and the cancer detection rate of 3.1% of all colonoscopies [10]. The rate of colorectal cancer detected by the program was 0.2%.

We currently aim to review 5-year results of the implementation of CRC screening program covering all the country with the purpose to inform quality improvement of the CRC screening program.

## 2. Materials and Methods

### 2.1. National Colorectal Cancer Screening Program in Lithuania

An organized screening program was initiated as a pilot project in 2009 in the two biggest cities of the country. From January 2014 the program covers all population aged 50–74 in Lithuania.

The program is divided into four services: (1) Information about the program (fecal immunochemical test (FIT) included), (2) Referral for colonoscopy, (3) Colonoscopy with or without biopsy, (4) Pathological examination and diagnosis. The program invites residents for FIT every 2 years. There are three FIT tests registered in Lithuania: AQ4 PolyCheck (Veda Lab, Alençon, France), IFOB (SureScreen Diagnostics Ltd., Derby, UK), and MediSmart (Lobeck Medical Ltd., Frick, Switzerland). There is no centrally organized invitation system in Lithuania. The general practitioner (GP) provides a written information leaflet. The FIT kit is given to the patient. The service is concluded when the results of FIT are reviewed. Subjects who are test-positive are referred and registered by GP for colonoscopy. If FIT is negative, the test is repeated in 2 years. Colonoscopy is performed under sedation by anesthetist with Midazolam. The program is described in more detail in the report of initial results of the CRC screening program [10]. The program is funded by National Health Insurance Fund (NHIF) and monitored by the program steering committee, consisting of representatives of surgeons, endoscopy specialists, epidemiologists, primary care physicians and pathologists, as well as representatives of the NHIF and the Ministry of Health.

### 2.2. Data Sources

The health system in Lithuania is mainly funded through the NHIF, which virtually covers the entire resident population. All residents of Lithuania are obliged to obtain health insurance coverage and the majority of them (98%) belong to public health insurance compensated by the NHIF [11]. The NHIF database was established in 1999 and contains demographic data and entries on the primary and secondary healthcare services, emergency and hospital admissions, and prescriptions of reimbursed medications for chronic diseases. For this analysis, we extracted the following data from the NHIF database between the 1 January 2014 and the 31 December 2018: number of individuals that belong to a target-screening cohort in each screening round (i.e., 50–74 years old) and available information on CRC screening services as indicated by specific codes (test-negative, test-positive, colonoscopy, biopsy results). Exact date and type of all provided services, test results, date and results of biopsy and histopathological examination were extracted together with the vital status at the end of follow-up for all men and women born between 1935 and 1968.

### 2.3. Data Analysis

Target populations were calculated for every screening year, number of screened persons, test results, and number of detected adenomas and cancer cases. Main performance indicators and outcome variables of CRC screening program were calculated as follows: screening coverage, proportion of test-positives and test-negatives, proportion of adenomas and cancer among screened patients and among test-positives, also detailed results of pathological examination are presented.

## 3. Results

In total, 1,523,109 men and women belonged to the target age group of screening (born between the years 1935 and 1968) and 754,061 (49.5%) individuals in the entire target population were screened for CRC cancer at least once during study period. During the study period only 357,403 persons participated in the program more than once and 43,409 than two times.

The main performance indicators of the CRC cancer screening program in Lithuania between 2014 and 2018 are presented in Table 1. Participation rate varied from 16.0% to 18.1% per year and was higher among women than among men. Proportion of test-positive and test-negative results was similar during all the study period—8.7% and 91.3% annually (Table 2). Colonoscopy was performed for 48.7–63.1% of test-positive patients, and there was a tendency to increased sedation use for colonoscopy (from 63.3% to 75.1%). Between 9.2% and 13.5% of test-positive patients received a biopsy of which 52.3–61.8% were positive for colorectal adenoma and 4.6–7.3% for colorectal carcinoma. Adenoma detection rate was from 18.9% to 21.41% for the colonoscopies performed within the program. CRC detection rate among test-positive individuals varied between 0.93% and 1.28%. Colorectal carcinoma was diagnosed for 1091 patients within CRC screening program between 2014 and 2018. The results of the pathological examinations as a percentage of the total number of biopsies performed are presented in Table 3. During study period, adenoma was found in 57.8% of biopsies, polyps and high-grade dysplasia in 15.4% and 13.9% of biopsies.

## 4. Discussion

We found, that 49.6% of screening-age individuals underwent at least one screening test within the CRC screening program during the period 2014–2018. There was a slight increase in screening program coverage comparing to our initial results (from 46% to 49.6%) [10]. CRC detection rate among test-positive individuals varied between 0.93% and 1.28% compared to the initial 0.2%. Adenomas were detected for 11.3% of FIT-positive persons. Adenoma detection rate for colonoscopy varied between 18.9% and 22.41%. High-grade dysplasia was found in 13.9% of all the biopsies compared to 11.8% previously.

Many organizational aspects influence the quality and effectiveness of the CRC screening program. The coverage and uptake are organizational parameters that have substantial impact on potential effectiveness of CRC screening program. In our study only half of the target group were covered by screening test at least once during the 5-year period. European and the US guidelines recommend a minimum CRC screening coverage of 65% and 60%, respectively. In addition, a threshold of 45% is acceptable in Europe [12,13]. In the review of the results and strategies of different screening programs worldwide participation in screening has varied greatly among different programs. The Netherlands showed the highest participation rate (68.2%) and some areas of Canada showed the lowest (16%) [14].

In EU countries, CRC screening program coverage varies between 10% and 71% [15]. Utilization of fecal tests and colonoscopy among people aged 50–74 years and the factors associated with uptake by type of screening offer differs across Europe and depends on the strategy of the program. The highest utilization of either test has been observed for countries with fully rolled out organized programs with fecal tests (ranging from 29.7% in Croatia to 66.7% in the UK) and countries offering both fecal tests and colonoscopy (from 22.7% in Greece to 70.9% in Germany) [15]. We found a higher number of female participation in the program. Participation rates were higher among women in programs that used the FIT test in other countries [14]. This can be explained that females traditionally have a stronger awareness of health and compliance with authority compared to male subjects.

Worldwide, most screening programs use a mailed patient contact strategy, a mailed contact plus screening kit strategy, or an office-visit contact strategy to encourage screening [14,16]. In Lithuania, a central screening registry and active invitation system has not been created and persons of eligible age are invited to participate in the screening program on opportunistic basis, therefore, coverage depends on the information provided by the GP. A historic meta-analysis concluded that the attendance rate depends on knowledge of cancer and screening provided by the GP [17]. It is generally low because of a feeling of embarrassment, a fear of screening complications or discomfort, a lack of communication with physicians and a lack of symptoms and awareness. Most studies have shown the strong correlation between the compliance with the program and the way the primary information was provided by the GP [18,19,20,21]. GP endorsement is a very important as this requires no effort for the participants. This is clearly seen in the CRC screening program in England, which showed an increase in participation if the invitation letter was added with a GP endorsement banner [22]. In a small pilot study by Tinmouth et al. the authors showed the importance of the family physician when providing information about the program [23]. Most studies show a large increase in participation when the FIT sample kit is included with the invitation [24,25,26]. Multiple steps of the other national screening programs are subject to sociodemographic inequalities, for example in Denmark males, individuals aged 60+ years and individuals who do not visit their GP regularly are at lower uptake and at a higher risk of being FIT- and/or CRC-positive [27].

Multiple themes which prevent or encourage participation in the CRC screening have been identified in the recent systematic review: psychology (fear of cancer), religion (believing cancer is the will of God), logistics (not knowing how to conduct the test), health-related factors (mental health), knowledge and awareness (lack of knowledge about the test), role of the general practitioner (being supported in taking the test by the general practitioner), and environmental factors (knowing someone who has participated in a screening program) [28]. Sending FIT kit with SMS reminders produced an absolute 17.7% increase in FIT kit return (*p* < 0.001) compared to sending just SMs reminders [29]. Sending nonparticipants, a reminder moderately increased participating rates from 41% to 45% in the English flexible sigmoidoscopy program [30]. The use of commitment device (patient self-ordering fecal immunochemical test (FIT) kits 3.8 times increases the odds of completing a kit compared to standard CRC screening participants (odds ratio, 3.77; 95% confidence interval, 3.57–3.98) [31]. Financial incentives (USD 10 guaranteed or USD 50 lottery) seem to increase the rate of participation and completion of screening: 95% of those who recalled the offer were screened compared to only 25% among those who did not remember the offer [32].

The important CRC screening variables include number of inadequate tests, number of positive tests that leads to referral to follow-up colonoscopy and referrals to follow-up colonoscopy [12]. Rates of positive screening test results reflects cut-off level chosen for adopted test. Quantitative FIT provides a possibility to customize cut-off level. In EU Member States, the FIT test range of positive rates in population-based studies was 4.4–11.1% in the first round, with one study reporting a rate in subsequent rounds of 3.9% [12]. Test-positive rates in our study varied between 7.33% and 10.24%, which is similar to that reported in other population-based studies.

Average compliance with colonoscopy among patients with a positive screening test in the randomized controlled trials using FIT ranges from 73% to 95%. Colonoscopy compliance rates range from 88% to 92% in population programs, while 90% rates of referral to follow-up colonoscopy for people with a positive screening test are acceptable (>95% is desirable) [13]. Compliance to colonoscopy among FIT-positive patients in Lithuania during the study period varied between 48.7% and 63.1%. This proportion is characteristic to non-population-based CRC screening programs in Europe and could reflect lack of referral system for screening test-positives to assessment colonoscopy.

Unlike GPs, the main role of the endoscopist in the screening process is to perform a high-quality colonoscopy. Few quality assurance measures for colonoscopy are known. One is the ileum intubation percentage—which should be >95% according to the European guidelines [9]. The adenoma detection rate is another important highly variable measure of the quality of mucosal inspection during colonoscopy [33]. Two large studies have validated the adenoma detection rate as a predictor of cancer prevention by colonoscopy [34,35]. These are the measures for choosing the endoscopist to participate in screening program. In our study, the adenoma detection rate (ADR) is high and it increased from 52.3% in 2014 to 61.8% in 2018. Compared to other studies the ADR ranged from 15% to 55% [36,37,38,39,40,41].

Moreover, the quality of pathology reporting seems to be acceptable as the percentage of high-grade dysplasia is reported to be 13.8% of the biopsies. The increased percentage (over recommended 10%) may be due to the fact that, statistically, only the highest degree of pathology is marked, so even in the presence of several adenomas only the highest-graded is marked in the statistical form and thus represented. Another possible explanation might be loss of small polyps during the colonoscopy without a pathological evaluation. CRC detection rate among test-positive individuals varied between 0.93% and 1.28%. This is lower compared to similar published studies: 4% and later 1% in Ireland [38], 6.2% in Slovenia [42], 5.5% in Korea [43].

The strength of our study is that it is a report of a population-based intervention with centralized registration, covering a whole country. The screening performance rates reported in this study will be of great importance when it comes to improving screening program organization and performance. The study also has some limitations that should be considered when interpreting its findings. First, the data were mostly collected for administrative purposes, therefore we had ability to report only selected indicators of program performance and outcomes. Information from the NHIF database also may have led to reporting biases and therefore the over- or underreporting of test use and the outcomes in this analysis. The data did not allow discrimination between screening and diagnostic FIT tests and colonoscopies. The NHIF database does not collect information on detected cancer stage and for this study, we had no possibility to link NHIF and Cancer registry records. Finally, data was not validated against medical records, however, an assessment performed by independent European experts in 2019 identified NHIF data as high quality [44].

In Lithuania a central screening registry and active invitation system has not been created and persons of eligible age are invited to participate in screening program on opportunistic basis, therefore, coverage depends on the information provided by the GP. A historic meta-analysis concluded that the attendance rate depends on knowledge of cancer and screening provided by the GP. The system of CRC screening in Lithuania in the time period presented is a compromise between national active and completely opportunistic systems. There is a central registration, complete reimbursement of fees and an obligation on primary care providers to invite the population of the screening age (50–74 years) for FIT test. There are also financial incentives for primary care providers, who recruit the highest proportion of their screen-eligible patients into the screening program. However, such system lacks the benefits of invitation with FIT kits included, adequate colonoscopy center allocation, and quality assurance of the centralized system. With the current analysis, it is evident, that changes in the invitation system need to be implemented, and the Ministry of Health initiated the changes to the screening system, which should be piloted from the third quarter of 2021 and which should roll out nationally from January 2022.

## 5. Conclusions

The screening program was able to screen 49% of all screening-age population at least once during a 5-year period. A screening database is needed to systematically evaluate the impact and performance of the national CRC screening program and quality assurance within the program.

## Figures and Tables

**Table 1 cancers-13-01129-t001:** Main performance indicators of the colorectal cancer screening program in Lithuania between 2014 and 2018.

Variable	2014	2015	2016	2017	2018
Target population (total, 50–74 years old)	1,297,155	1,311,156	1,325,082	1,336,298	1,346,670
Individuals screened	234,881	210,036	234,185	229,344	241,553
Coverage (participation rate, %)	18.1	16.0	17.7	17.2	17.9
Target population (male, 50–74 years old)	633,764	643,646	653,097	661,051	669,042
Individuals screened	88,425	84,024	93,289	90,101	96,971
Coverage (participation rate, %)	14.0	13.1	14.3	13.6	14.5
Target population (female, 50–74 years old)	663,391	667,510	671,984	675,247	677,628
Individuals screened	146,456	126,012	140,896	139,243	144,582
Coverage (participation rate, %)	22.1	18.9	21.0	20.6	21.3

**Table 2 cancers-13-01129-t002:** Performance indicators for FIT and colonoscopy and outcomes of colorectal cancer screening program in Lithuania between 2014 and 2018.

Variable	2014	2015	2016	2017	2018
Target population	1,297,155	1,311,156	1,325,082	1,336,298	1,346,670
FIT results					
Number of FIT negative persons (% of all test)	217,659	188,529	211,424	210,137	222,394
	(92.67)	(89.76)	(90.28)	(91.63)	(92.07)
Number of FIT positive persons (% of all test)	17,222	21,507	22,761	19,207	19,159
	(7.33)	(10.24)	(9.72)	(8.37)	(7.93)
Compliance to colonoscopy (% of FIT positive)	8382	12,630	12,916	12,112	11,140
	(48.7)	(58.7)	(56.8)	(63.1)	(58.2)
Colonoscopy with sedation (% of all colonoscopies)	5304	7993	9575	8907	8557
	(63.3)	(63.3)	(74.1)	(73.5)	(75.1)
Colonoscopy without sedation (% of all colonoscopies)	3078	4637	3341	3205	2583
	(36.7)	(36.7)	(25.9)	(26.5)	(24.9)
Number of patients with biopsies performed	3029	4519	4717	4316	3499
Number of patients with detected adenoma	1584	2564	2895	2601	2163
Adenoma detection rate (% of biopsies)	18.9	20.3	22.41	21.47	19.42
Proportion of adenoma of test-positive persons (%)	9.20	11.92	12.72	13.54	11.29
Proportion of adenoma of screened persons (%)	0.67	1.22	1.24	1.13	0.90
Number of persons with detected CRC	221	248	217	227	178
Positive predictive value for CRC	7.30	5.49	4.60	5.26	5.09
% CRC of test-positive	1.28	1.15	0.95	1.18	0.93
% CRC of screened persons	0.09	0.12	0.09	0.10	0.07

CRC—colorectal cancer; FIT—fecal immunochemical test.

**Table 3 cancers-13-01129-t003:** Results of the pathological examination of the colorectal cancer screening program in Lithuania between 2014 and 2018.

Pathology Diagnoses	*N*	%
Poor-quality specimen	76	0.28
Normal tissue	1756	6.42
Polyp	4399	16.09
Adenoma	15,653	57.25
High-grade dysplasia	3368	12.32
IBD-associated neoplasia	195	0.71
Carcinoma	1766	6.46
Neuroendocrine tumor	20	0.07
Benign nonepithelial tumor	84	0.31
Malignant nonepithelial tumor	11	0.04
Lymphoma	6	0.02
Secondary tumor	7	0.03
Overall	27,341	100

IBD—inflammatory bowel disease.

## Data Availability

Data is contained within the article. The data presented in this study are available in the article.

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
