# Peer review of "National Colorectal Cancer Screening Program in Lithuania: Description of the 5-Year Performance on Population Level"

_cancers, 2021, doi:10.3390/cancers13051129_

Round 1

Reviewer 1 Report

This paper does not describe the results of a population-based screening program, as recruitment into the program was based upon referral by GPs.  A true population-based screening program would base selection upon a population-based data base.

Therefore this paper is of limited general interest, it is simply a report documenting those who had colorectal cancer or its precursors detected.  It may be useful to the authorities in Lithuania who fund the program, but is of extremely low general interest.

Author Response

Dear Reviewer,

Thank you for your letter and constructive comments concerning our manuscript entitled “National Colorectal Cancer Screening Program in Lithuania: description of the 5-year performance on population level”. The paper was revised substantially. Following changes have been made. They are as follows:

Revised paragraphs, sentences, words are below:

This paper does not describe the results of a population-based screening program, as recruitment into the program was based upon referral by GPs.  A true population-based screening program would base selection upon a population-based data base.

Therefore this paper is of limited general interest, it is simply a report documenting those who had colorectal cancer or its precursors detected.  It may be useful to the authorities in Lithuania who fund the program, but is of extremely low general interest.

We strongly disagree with the statement of the reviewer. As colorectal cancer screening remains debatable among professional communities (physicians and public health experts) (https://pubmed.ncbi.nlm.nih.gov/26331705/ ), regarding testing modality and organizational issues, we believe, that reporting of performance results of colorectal cancer screening in national cohorts contributes to better outcomes of colorectal cancer patients.

More references:

  1. https://pubmed.ncbi.nlm.nih.gov/28507758/
  2. https://pubmed.ncbi.nlm.nih.gov/25734239/
  3. https://www.thelancet.com/journals/lancet/article/PIIS0140-6736(05)62856-5/fulltext

Thank you very much indeed.

Yours sincerely

Reviewer 2 Report

This is an informative article on the evaluation of a national colorectal cancer screening program that public health practitioners would find useful. I have some minor comments below that may help to strengthen this manuscript. 

Introduction: Page 2, line 48: The figure of 700,000 new cases of colorectal cancer worldwide seems too low, so you may want to double check this on the GLOBOCAN website. 

line 69: You may want to give some more details here about the purpose of the study, such as "this was a descriptive study to inform quality improvement of the CRC screening program..."

Methods: Page 2, line 84: Did this project require Institutional Review Board approval, or was it considered exempt research?

Page 3, line 99: what software was used for analysis? 

Table 2: Line with "Number of patients with biopsies performed (% of test-positive)" - did you intend to add percentages here as well?

Page 4, line 142:  Are any symptoms of colorectal cancer (e.g. anemia, blood in the stool, changes in bowel habits, etc.) recorded at the time of screening that may indicate a diagnostic test vs. a screening test? I wonder how often clinicians refer patients to the screening program because of suspected colorectal cancer?

Page 5, line 193: How does the referral process typically work in Lithuania? Does the general practitioner provide a referral to a gastroenterology clinic, or is it dependent on the patient to find a gastroenterologist and make an appointment? I think it would be helpful to readers to describe the typical process of care. 

Page 6, line 221: I think it would be helpful to readers to elaborate more on potential reporting biases. Are these potentially due to differences in codes used for payment or not capturing all services rendered? Have the data ever been validated against medical records? 

Author Response

Dear Reviewer,

Thank you for your letter and constructive comments concerning our manuscript entitled “National Colorectal Cancer Screening Program in Lithuania: description of the 5-year performance on population level”. The paper was revised substantially. Following changes have been made. They are as follows:

Revised paragraphs, sentences, words are below:

This is an informative article on the evaluation of a national colorectal cancer screening program that public health practitioners would find useful. I have some minor comments below that may help to strengthen this manuscript. 

Introduction: Page 2, line 48: The figure of 700,000 new cases of colorectal cancer worldwide seems too low, so you may want to double check this on the GLOBOCAN  website.    

Thank you for your comment – corrected.

line 69: You may want to give some more details here about the purpose of the study, such as "this was a descriptive study to inform quality improvement of the CRC screening program..."

The purpose described more clearly.

Methods: Page 2, line 84: Did this project require Institutional Review Board approval, or was it considered exempt research?

Ethical approval number was added to the Methods part.

Page 3, line 99: what software was used for analysis? 

For data management we used standard MS programs MS Access and MS Excel, therefore we not mentioned the software in Methods section. No special software was used for analysis.

Table 2: Line with "Number of patients with biopsies performed (% of test-positive)" - did you intend to add percentages here as well?

Thank you for the comment. The mistake was corrected.

Page 4, line 142:  Are any symptoms of colorectal cancer (e.g. anemia, blood in the stool, changes in bowel habits, etc.) recorded at the time of screening that may indicate a diagnostic test vs. a screening test? I wonder how often clinicians refer patients to the screening program because of suspected colorectal cancer?

We agree that this would be a very interesting data; however, we do not have it. In addition, this is slightly of our topic.

Page 5, line 193: How does the referral process typically work in Lithuania? Does the general practitioner provide a referral to a gastroenterology clinic, or is it dependent on the patient to find a gastroenterologist and make an appointment? I think it would be helpful to readers to describe the typical process of care. 

The pathway of the screenee was added in more details in METHODS part. (briefly the screened person is referred and registered for the colonoscopy by GP)

Page 6, line 221: I think it would be helpful to readers to elaborate more on potential reporting biases. Are these potentially due to differences in codes used for payment or not capturing all services rendered? Have the data ever been validated against medical records? 

The national health insurance fund database includes all patients, who underwent CRC screening in the country as a part of national health service. It may be possible, that some paid services in the private practice, such as voluntary FIT or screening colonoscopy are not included, but they are extremely rare and account for less than 0,1% of all screening services. The data on these services is not covered in the database. National health insurance fund routinely and randomly tests the quality of the registration of the screening services and the validity of medical documentation and the discrepancies result in change in the numbers reported in the database.

More limitations added in Limitation part.

Thank you very much indeed.

Yours sincerely

Reviewer 3 Report

I recommend that you should describe the population-based CRC cancer screening program in Lithuania more detail than now. (FOBT methods used FIT test with three commercial kits. FIT is performed in home test. The service of referral for colonoscopy is provided by the primary care physician if the FIT is positive. The information service is reimbursed for everyone once every 2 years and if the FIT result is negative, a repeat test should be performed after 2 years. etc) I think that most readers could understand your country’s screening program with this manuscript without your previous work.

You'd better show the brief enrolled participants by study flow diagram.  

The results were showed by table. However, it could not easily recognize or understand the main outcomes in table? How about do you present the key results by graphs?

In your data, you reported the annual low participant rate in Lithuania. Is there any national policy or methods for improving or raising the participants by government? And you’d be better discussed about other references of raising the participants rates. And, how many persons did participate the screening program more than two times during study period? 

You used NHIF or national data of CRC screening program. Could you show us the stage of CRC detected by screening program comparing to those of non-screened patients (or comparing to those of CRC patients diagnosed before cancer screening program started)? Is there any change of stage in CRC by screening program?

In previous manuscript, your fecal tests were FIT (fecal immunochemical test). I would recommend the fecal tests (iFOBT)were unified with FIT.

You describe the ‘colonoscopy under/without general anesthesia’. Word of general anesthesia brings misunderstanding to reader as an anesthetic operation. How about does change the word from general anesthesia to sedation?

The context of the second paragraph in introduction is not smooth, so it would be better to rewrite the second paragraph.

Author Response

Dear Reviewer,

Thank you for your letter and constructive comments concerning our manuscript entitled “National Colorectal Cancer Screening Program in Lithuania: description of the 5-year performance on population level”. The paper was revised substantially. Following changes have been made. They are as follows:

Revised paragraphs, sentences, words are below:

I recommend that you should describe the population-based CRC cancer screening program in Lithuania more detail than now. (FOBT methods used FIT test with three commercial kits. FIT is performed in home test. The service of referral for colonoscopy is provided by the primary care physician if the FIT is positive. The information service is reimbursed for everyone once every 2 years and if the FIT result is negative, a repeat test should be performed after 2 years. etc) I think that most readers could understand your country’s screening program with this manuscript without your previous work.

The pathway for the screenee was described in more details in METHODS part.

You'd better show the brief enrolled participants by study flow diagram.

Thank you for the suggestion. Indeed, we tried to make it as a flow diagram; however, it does not provide any additional value and information.

The results were showed by table. However, it could not easily recognize or understand the main outcomes in table? How about do you present the key results by graphs?

Thank you for the suggestion. However, after discussion with other co-authors we preferred to show main results in Tables, as it is.

In your data, you reported the annual low participant rate in Lithuania. Is there any national policy or methods for improving or raising the participants by government? And you’d be better discussed about other references of raising the participants rates. And, how many persons did participate the screening program more than two times during study period? 

In Lithuania central screening registry and active invitation system not created and per-sons of eligible age are invited to participate in screening program on opportunistic basis, therefore, coverage depends on the information provided by the GP. A historic meta-analysis concluded that the attendance rate depends on knowledge of cancer and screening provided by the GP.

The methods of improving the participants are discussed in the Discussion (added the paragraph with new references).

You used NHIF or national data of CRC screening program. Could you show us the stage of CRC detected by screening program comparing to those of non-screened patients (or comparing to those of CRC patients diagnosed before cancer screening program started)? Is there any change of stage in CRC by screening program?

Unfortunately, NHIF database does not collect information on detected cancer stage and for this study we had no possibility to link NHIF and Cancer registry records. This is mentioned as study limitation yet.

In previous manuscript, your fecal tests were FIT (fecal immunochemical test). I would recommend the fecal tests (iFOBT)were unified with FIT.

It was unified as suggested.

You describe the ‘colonoscopy under/without general anesthesia’. Word of general anesthesia brings misunderstanding to reader as an anesthetic operation. How about does change the word from general anesthesia to sedation?

We totally agree with you. “General” changed to “sedation”

The context of the second paragraph in introduction is not smooth, so it would be better to rewrite the second paragraph.                                                              The paragraph was rewritten.

Thank you very much indeed.

Yours sincerely

Reviewer 4 Report

Dulskas et al report on the results of the colorectal cancer screening programme in Lithuania. The data are well organized and the manuscript has been written in a sufficient style. However, English language needs to be revised. This is an important report that merits publication.

I have the following comments: 

  1. It is reported that the majority of colonoscopies is done in general anesthesia. Please provide information on what medication is given, disoprivan, midazolam, other? Who provides the medication in Lithuania, the nurse, the endoscopist, an anesthesiologist?
  2. 1091 colorectal cancers were diagnosed during the reporting period. Please provide TNM/UICC stages. Is the majority of cancers early stage?
  3. Please provide more detailed information on the respective number of screened individuals who participated once, twice, three times etc. in the screening. What is the number of positive tests in the respective groups?
  4. How do you define adenoma detection rate? "% of positive biopsies" is not the commonly used definition. Adenoma detection rate is defined as number of colonoscopies with detection of adenomas compared to the total number of colonoscopies. Please revise paper accordingly.

Author Response

Dear Reviewer,

Thank you for your letter and constructive comments concerning our manuscript entitled “National Colorectal Cancer Screening Program in Lithuania: description of the 5-year performance on population level”. The paper was revised substantially. Following changes have been made. They are as follows:

Revised paragraphs, sentences, words are below:

Dulskas et al report on the results of the colorectal cancer screening programme in Lithuania. The data are well organized and the manuscript has been written in a sufficient style. However, English language needs to be revised. This is an important report that merits publication.

I have the following comments: 

  1. It is reported that the majority of colonoscopies is done in general anesthesia. Please provide information on what medication is given, disoprivan, midazolam, other? Who provides the medication in Lithuania, the nurse, the endoscopist, an anesthesiologist?

This information was added to METHODS part – briefly patients are sedated by anaesthetist with midazolam.

  1. 1091 colorectal cancers were diagnosed during the reporting period. Please provide TNM/UICC stages. Is the majority of cancers early stage?

Unfortunately, NHIF database not collects information on detected cancer stage and for this study we had no possibility to link NHIF and Cancer registry records. This is mentioned as study limitation yet.

  1. Please provide more detailed information on the respective number of screened individuals who participated once, twice, three times etc. in the screening. What is the number of positive tests in the respective groups?

During study period only 357,403 persons participated in the program more than once and 43,409 than two times. We added this sentence to the Results section.

  1. How do you define adenoma detection rate? "% of positive biopsies" is not the commonly used definition. Adenoma detection rate is defined as number of colonoscopies with detection of adenomas compared to the total number of colonoscopies. Please revise paper accordingly.

Thank you for your comment. We have corrected the mistake and revised the paper accordingly.    

Thank you very much indeed.

Yours sincerely,

Audrius Dulskas, MD, PhD